# Autophosphorylation and Self-Activation of DNA-Dependent Protein Kinase

**DOI:** 10.3390/genes12071091

**Published:** 2021-07-19

**Authors:** Aya Kurosawa

**Affiliations:** 1Graduate School of Science and Engineering, Gunma University, Kiryu 376-8515, Japan; aya.kurosawa@gunma-u.ac.jp; Tel.: +81-277-30-1445; 2Food and Health Science Education and Research Center, Gunma University, Kiryu 376-8515, Japan

**Keywords:** DNA-dependent protein kinase, autophosphorylation, non-homologous end-joining, differentiation

## Abstract

The DNA-dependent protein kinase catalytic subunit (DNA-PKcs), a member of the phosphatidylinositol 3-kinase-related kinase family, phosphorylates serine and threonine residues of substrate proteins in the presence of the Ku complex and double-stranded DNA. Although it has been established that DNA-PKcs is involved in non-homologous end-joining, a DNA double-strand break repair pathway, the mechanisms underlying DNA-PKcs activation are not fully understood. Nevertheless, the findings of numerous in vitro and in vivo studies have indicated that DNA-PKcs contains two autophosphorylation clusters, PQR and ABCDE, as well as several autophosphorylation sites and conformational changes associated with autophosphorylation of DNA-PKcs are important for self-activation. Consistent with these features, an analysis of transgenic mice has shown that the phenotypes of DNA-PKcs autophosphorylation mutations are significantly different from those of DNA-PKcs kinase-dead mutations, thereby indicating the importance of DNA-PKcs autophosphorylation in differentiation and development. Furthermore, there has been notable progress in the high-resolution analysis of the conformation of DNA-PKcs, which has enabled us to gain a visual insight into the steps leading to DNA-PKcs activation. This review summarizes the current progress in the activation of DNA-PKcs, focusing in particular on autophosphorylation of this kinase.

## 1. Introduction

Cells regulate the cell cycle and gene expression rapidly in response to both intracellular and extracellular stresses, during which signal transduction is primarily regulated via phosphorylation [1,2,3]. Within these signal transduction pathways, members of the PI3K-related kinase (PIKK) family, including ataxia-telangiectasia mutated (ATM), ataxia telangiectasia and Rad3-related (ATR), the DNA-dependent protein kinase catalytic subunit (DNA-PKcs), the mammalian target of rapamycin (mTOR), the suppressor with morphological effect on genitalia-1 (SMG-1) and the TRAF and TNF receptor-associated protein (TTRAP), are known to play central roles [4]. Although TTRAP lacks serine/threonine kinase activity, all of these proteins are characterized by the presence of HEAT, FAT and kinase domains [5,6,7]. Phylogenetic analyses of DNA-PKcs have recently indicated that many metazoans and plants have putative gene encoding DNA-PKcs. However, many model organisms such as *Saccharomyces cerevisiae*, *Schizosaccharomyces pombe*, *Caenorhabditis elegans*, *Drosophila melanogaster* and *Arabidopsis thaliana* lack an ortholog of DNA-PKcs. Therefore, DNA-PKcs appears to have been lost or its functions may have been diversified as organisms adapted to their environment during the course of evolution [8]. DNA-PKcs has been established as playing a central role in non-homologous end-joining (NHEJ), a DNA double-strand break (DSB) repair pathway, and its activation is dependent on its interactions with the Ku complex [9,10,11]. DNA-PKcs binds rapidly to DNA in the presence of Ku and it has been reported that the interaction of Akt1 and/or EGFR with DNA-PKcs promotes the activation of DNA-PK [12]. During NHEJ, the Ku70/Ku80 complex rapidly binds to DNA ends in response to DSBs; DNA-PKcs then binds to these DNA ends followed by the recruitment of NHEJ factors such as Artemis and DNA polymerase λ/μ, which subsequently process the DNA ends to generate ligatable ends [11,13]. In organisms that lack the gene encoding for DNA-PKcs, such as fission yeast (*Schizosaccharomyces pombe*), the Mre11/Rad50/Xrs2 complex interacts with NHEJ factors such as DNA ligase IV, is involved in the end-resection step of NHEJ and ensures NHEJ fidelity [14,15,16]. It has also been suggested that PAXX interacts with the Ku complex and stabilizes DNA ends [10,17,18] and that XRCC4/XLF filaments are formed within the vicinity of DNA ends to protect the DNA ends from further degradation [10,19,20,21,22]. However, the order and timing of DNA-PKcs binding and recruitment of these NHEJ factors have yet to be sufficiently established.

NHEJ is also involved in V(D)J recombination, which is responsible for the diversity of antigen recognition sites in antibodies and T-cell receptors [23]. In particular, DNA-PKcs is essential for opening the hairpin DNA formed after cleavage by the RAG1/RAG2 complex [24]. Artemis, which is the enzyme responsible for opening the hairpin DNA, has endonuclease activity [11,24]. However, when it forms a complex with DNA-PKcs and undergoes phosphorylation by DNA-PKcs, it acquires structure-specific exonuclease activity [11,24]. Therefore, an abnormality in the gene encoding DNA-PKcs results in radiosensitive severe combined immunodeficiency (RS-SCID) [23].

It is known that DNA-PKcs undergoes autophosphorylation and phosphorylates other NHEJ factors including Ku70, Ku80, Artemis, PNKP and XRCC4 and the roles of DNA-PKcs substrates and DNA-PKcs autophosphorylation in the DSB repair process have been well analyzed [25]. For example, it has been reported that autophosphorylation of DNA-PKcs is involved in end-processing, the pathway choice for DSB repair and normal mitotic progression [26,27,28,29]. Moreover, the phosphorylation status of DNA-PKcs is associated with a conformational change, which is required for self-activation [30,31,32,33,34,35]. The three-dimensional structure of the Ku complex was reported in detail in 2001 [36]. A visualization of the three-dimensional structure of DNA-PKcs was attempted by cryo-electron microscopy (Cryo-EM) in 1998 although the resolution was low [37,38,39]. More recently, there have been a number of higher resolution structural characterizations of DNA-PKcs using X-ray and Cryo-EM analyses [40,41,42,43]. Based on this information, in this review, the current understanding of autophosphorylation and the activation of DNA-PKcs has been summarized.

## 2. Self-Activation of a DNA-Dependent Protein Kinase Catalytic Subunit

DNA-PKcs primarily recognizes serine/glutamine and threonine/glutamine motifs in substrate proteins and phosphorylate serine and threonine residues in the presence of the Ku complex and double-stranded DNA (Figure 1) [44]. As shown in Figure 2, DNA-PKcs consists of the HEAT, FAT and kinase domains. The HEAT domain contains two major autophosphorylation clusters referred to as the S2056 (S2023–S2056) and T2609 (T2609–T2647) clusters, also known as the PQR and ABCDE clusters, respectively [7,26,33].

Based on the Cryo-EM analysis, the conformations of DNA-PKcs can be classified into three major groups based on its binding to the Ku complex and DNA and its corresponding activation state: (i) the DNA-PKcs-DNA complex, (ii) the DNA-PK holoenzyme in the inactivated state and (iii) the DNA-PK holoenzyme in the activated state [40]. In the DNA-PKcs-DNA complex, it was found that DNA-PKcs binds to DNA alone, although weakly; this is consistent with the weak kinase activity of DNA-PKcs in the absence of Ku but in the presence of DNA [45,46]. In addition to forming a DNA-PK holoenzyme, the self-activation of DNA-PK requires the physical interaction between Ku80 and DNA-PKcs as well as conformational changes in the FAT and kinase domains of DNA-PKcs. When the DNA-PK holoenzyme is in the activated state, a putative inositol 6-phosphate (IP6) binding site appears [40]. IP6 was reported by Hanakahi and West in 2000 as a factor that promotes end-joining in vitro [47]. Interestingly, IP6 binds to the Ku complex rather than to DNA-PKcs [24,48]. Consistent with this, it has been suggested that DNA-PKcs lacks an IP6 binding site unlike other PI3KKs [49]. To verify this contradiction, it is necessary to assess in detail whether IP6 binds to DNA-PKcs or the Ku complex.

With respect to autophosphorylation, there is yet to be a comprehensive characterization of PQR clusters (S2029, S2041, S2053 and S2056 in humans and S2026, S2038, S2050 and S2053 in mice), including the locations and structural changes associated with phosphorylation [40]. Of these phosphorylation sites, autophosphorylation of S2056 has been well analyzed. S2056 is phosphorylated in trans, at least in vitro, in response to ionizing radiation (IR) [50]. Phosphorylation of S2056 influences the pathway choice by restricting the end-processing by Artemis during the NHEJ reaction. It has been reported that the autophosphorylation efficiency of S2056 is related to the balance with O-linked β-N-acetylglucosamine (O-GlcNAc) modification, a post-translational modification of proteins, and the interaction of DNA-PKcs with casein kinase II and/or Akt1 [12,51,52]. However, even though all phosphorylation sites in the PQR cluster are replaced with alanine (hereafter referred to as DNA-PKcs^PQR^), phosphorylation of KAP-1 by DNA-PKcs occurs normally in response to IR. Furthermore, mouse embryonic fibroblasts (MEFs) and B-cells derived from *DNA-PKcs^PQR/PQR^* mice exhibited a mild sensitivity to IR [53]. Collectively, these findings indicate that phosphorylation at the PQR cluster is dispensable for the activation of DNA-PK.

In contrast to the PQR cluster, the ABCDE cluster plays an important role in the self-activation of DNA-PKcs [30,33,54]. Within the ABCDE cluster, six (in humans) or five (in mouse) serine and threonine residues (T2609, S2612, T2620, S2624, T2638 and T2647 in humans and T2605, S2614, S2616, T2634 and T2643 in mice) undergo phosphorylation in vitro. Four of these amino acids (T2609, S2612, T2638 and T2647) have also been found to undergo phosphorylation in vivo [32]. Mammalian cells and transgenic mice expressing DNA-PKcs with all phosphorylation sites or five (excluding the non-S/Q motif S2645 in all ABCDE clusters) or three threonine residues (T2609, T2638 and T2647 in humans; T2605, T2634 and T2463 in mice) of the ABCDE cluster replaced with alanine (hereinafter referred to as DNA-PKcs^6A^, DNA-PKcs^5A^ and DNA-PKcs^3A^, respectively) are often used to analyze the role of the ABCDE cluster [30,31,32,34,35]. For example, analyses using V3 CHO cells expressing human DNA-PKcs^6A^ and in vitro NHEJ assays using autophosphorylation mutant DNA-PKcs proteins revealed that DNA-PKcs remains bound to DNA ends in a Ku-dependent manner and that the NHEJ reaction does not proceed any further if the ABCDE cluster does not undergo phosphorylation [30,32]. Interestingly, two distinct types of conformational changes associated with phosphorylation have been reported in the ABCDE cluster. Briefly, in one scenario, the ABCDE cluster extends from within the HEAT domain to the DNA binding site and autophosphorylation at the ABCDE cluster occurs in trans. An alternative view suggests that the ABCDE cluster is located externally to the HEAT domain, adjacent to the substrate binding groove of the kinase domain, and that autophosphorylation at the ABCDE cluster occurs in cis (Figure 3) [32,40,55]. Consequently, it would be desirable to verify the precise location of the ABCDE cluster in future studies.

In addition to these clusters, it has been reported that autophosphorylation of the N-terminal cluster (S56 and S72) is involved in the inactivation of DNA-PKcs. In particular, phosphorylation of S72 has been suggested to destabilize the binding of DNA-PKs to DNA [28,40,56]. It has also been suggested that phosphorylation of the JK clusters (T946 and S1004) does not affect enzyme activity but is involved in the pathway choice [56]. Phosphorylation of the leucine zipper clusters (S1470 and S1546) and T1865 appears to have little effect on DNA-PK functions such as radiosensitivity, V(D)J recombination, the inhibition of homologous recombination and the assembly into Ku-bound DNA [56]. S3205 undergoes autophosphorylation modification by DNA-PKcs in vitro. However, it is phosphorylated by ATM in response to DNA damage in vivo [56]. Regarding T3950, as this residue is located within the activation loop of the kinase domain, its phosphorylation and dephosphorylation have been suggested to play a switching role with respect to kinase activity [40,42,57].

## 3. Effects of DNA-PKcs on Development and Differentiation Determined by Comparing Knockout and Mutant Mice

The importance of DNA-PKcs phosphorylation in the development of tissues and cells has been demonstrated using mouse models. For example, it has been found that while *DNA-PKcs^-/-^* mice develop normally, a DNA-PKcs kinase-dead mutant (*DNA-PKcs^kd/kd^*), in which the aspartic acid at position 3922 of mouse DNA-PKcs is replaced by alanine, is embryonic lethal [58,59,60]. A histological analysis of the E14.5 brains of these *DNA-PKcs^kd/kd^* mice revealed severe neuronal apoptosis, which was similar to that observed in *Xrcc4* knockout mice. Neuronal apoptosis is primarily detected in the post-mitotic intermediate zone, thereby indicating that apoptosis occurs during the G_0_/G_1_ phase of the cell cycle in which NHEJ repair is predominant [59]. In addition, an accumulation of chromosomal breaks was detected in ES cells and MEFs derived from *DNA-PKcs^kd/kd^* mice. Although ES cells derived from *DNA-PKcs^-/-^* were found to be only moderately sensitive to IR, those derived from *DNA-PKcs^kd/kd^* mice were notably more sensitive. Although DNA-PK is also implicated in the pathway choice, a homologous recombination repair in ES cells derived from *DNA-PKcs^kd/kd^* mice has been observed to be comparable with that in wild-type ES cells, thereby indicating that genomic instability caused by the kinase-dead mutation in DNA-PK is attributable to NHEJ defects. Although extrachromosomal end-ligation does not proceed in B-cells derived from *DNA-PKcs^kd/kd^* mice, this end-ligation failure can be rescued by Ku deletion, consequently indicating that unphosphorylated DNA-PKcs blocks ligation [59]. Considering the fact that *DNA-PKcs^PQR/PQR^* mice are healthy and the aforementioned importance of the ABCDE cluster in DNA-PK activation, it appears that the fetal lethality observed in *DNA-PK^kd/kd^* mice is associated with an aberrant phosphorylation of the ABCDE cluster. Moreover, *DNA-PKcs^3A/3A^* mice, wherein three amino acids in the ABCDE cluster are replaced with alanine, have been found to have bone marrow and telomere abnormalities and die approximately 10 days after birth [35,61]. Accordingly, this phenotypic difference between *DNA-PKcs^3A/3A^* and *DNA-PKcs^kd/kd^* mice would appear to indicate that in addition to phosphorylation of the ABCDE cluster, autophosphorylation of DNA-PKcs is an important process for the development (at least neurogenesis) of individual mice. Furthermore, given that the embryonic lethality characterizing *DNA-PK^kd/kd^* mice can be rescued by Ku deficiency, the effects of autophosphorylation such as those of S72 and T2950 (in human DNA-PKcs), which are required for the dissociation of DNA-PKcs from DNA, would not be negligible.

Gene knockout and kinase-dead mutations in DNA-PKcs also give rise to further phenotypic differences. Recently, it has been reported that DNA-PKcs is involved in rRNA processing and hematopoiesis in a Ku-dependent manner [34]. In mice, the loss of cNHEJ factors other than DNA-PKcs leads to the development of pro-B lymphoma in a TP53-deficient background [62,63,64], whereas, in the same background, most DNA-PKcs kinase-dead mutant mice die within 40 days without developing lymphoma [34]. An analysis of the bone marrow and spleen from *DNA-PKcs^kd/kd^ TP53^-/-^* mice revealed that these mice developed myelodysplastic syndrome. Studies focusing on autophosphorylation of DNA-PKcs have indicated that while *DNA-PKcs^PQR/PQR^* mice were healthy, *DNA-PKcs^5A/5A^* and *DNA-PKcs^3A/3A^* mice showed bone marrow failure [34,53]. The fact that Ku binds to both DNA and RNA [65,66], and that the Ku complex and DNA-PKcs localize in the nucleolus in the absence of DNA damage [67], suggests that DNA-PK may act in an RNA-dependent manner in the nucleolus. Indeed, an analysis using v-ABL kinase-transformed pro-B cell lines from *DNA-PKcs^kd/kd^* and *DNA-PKcs^3A/3A^* mice revealed that translation is reduced in these cells [34]. Furthermore, the findings of a ChIRP-MS analysis indicated that the DNA-PK holoenzyme binds to U3 small nucleolar RNA (snoRNA), which is involved in the maturation of the 40S ribosomal subunit. Further in vitro phosphorylation assays have revealed that DNA-PKcs binds to the snoRNA stem-loop structure in a Ku-dependent manner, thereby causing autophosphorylation of the ABCDE cluster. Accordingly, these observations indicate that *DNA-PKcs^kd/kd^*, *DNA-PKcs^5A/5A^* and *DNA-PKcs^3A/3A^* mice show Ku-dependent aberrant 18S rRNA processing, which results in an overall reduction in protein synthesis in hematopoietic cells [34]. However, although the role of the ABCDE cluster in rRNA processing remains to be conclusively established, Shao et al. speculated that the conformation of DNA-PKcs may differ depending on the identity of the binding nucleic acid (i.e., DNA or RNA) [34]. Indeed, an elucidation of the three-dimensional structure of DNA-PKcs bound to RNA is presumably the most important issue to be resolved regarding the role of DNA-PKcs in rRNA processing.

## 4. Conclusions

In this review, how the activation of DNA-PKcs is accompanied by autophosphorylation and conformational changes as well as the binding of IP6, the Ku complex and DNA have been described. The evidence also indicates that autophosphorylation of DNA-PKcs is important for protein synthesis in hematopoietic cells as well as in the development of neuronal and lymphoid cells. The findings reported herein highlight the continuing importance of studies based on animal models to clarify the roles of DNA-PKs. Although there have been notable advances in the structural analysis of macromolecular proteins by Cryo-EM, the structures of DNA-PKcs have yet to be sufficiently clarified. Consequently, the precise locations of the ABCDE and PQR clusters and the conformation of DNA-PKs in the ATP-bound state remain to be determined. In conclusion, it is envisaged that a complementary combination of cellular, molecular, biochemical and structural analyses will contribute to gaining a more comprehensive understanding of the mechanisms underlying DNA-PK activation, thereby facilitating a further elucidation of the multiple functions of DNA-PKcs.

## Figures and Tables

**Figure 1 genes-12-01091-f001:**
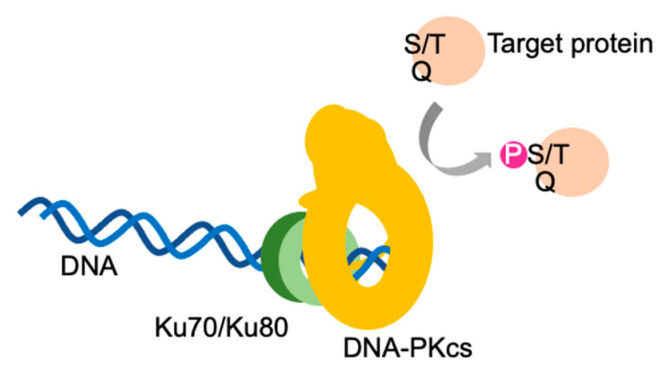
DNA-dependent protein kinase. DNA-PK holoenzyme consists of a DNA-dependent protein kinase catalytic subunit (DNA-PKcs), the Ku complex (Ku70/Ku80) and double-stranded DNA. DNA-PKcs acquires kinase activity in the presence of the Ku complex and double-stranded DNA. DNA-PK holoenzymes preferentially phosphorylate the serine (S) and threonine (T) of serine /glutamine (Q) and threonine/glutamine motifs in target proteins.

**Figure 2 genes-12-01091-f002:**
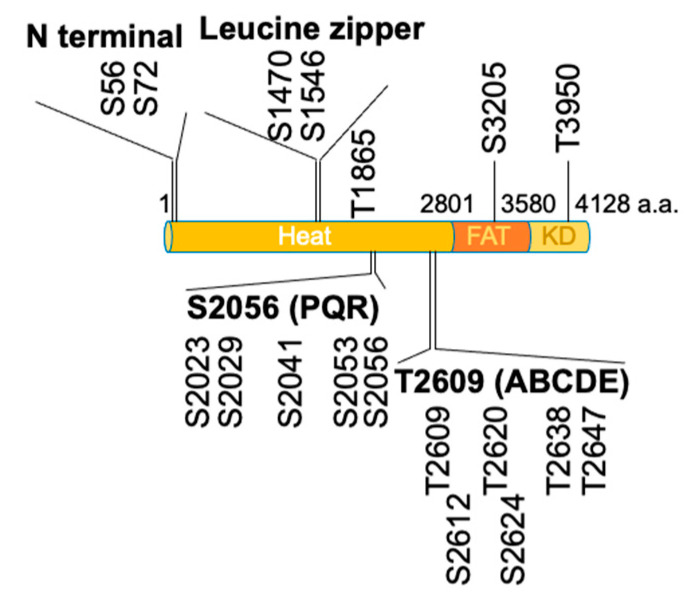
DNA-PKcs consists of the HEAT domain, FAT domain and kinase domain (KD). There are four autophosphorylation clusters (N terminal, leucine zipper, S2056/PQR and T2609/ABCDE) and one autophosphorylation site in the HEAT domain. There is an autophosphorylation site in the FAT (S3205) and KD (T3950) domains, respectively.

**Figure 3 genes-12-01091-f003:**
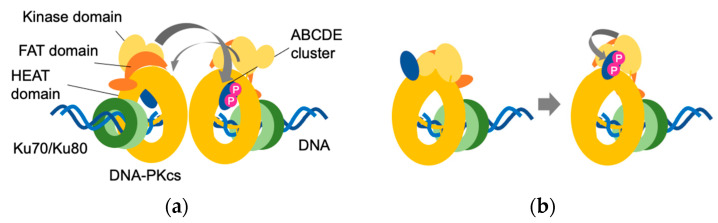
Schematic representation of a putative ABCDE cluster. (**a**) A putative ABCDE cluster is located inside the HEAT domain. Autophosphorylation occurs in trans; (**b**) Another putative ABCDE cluster is located outside the HEAT domain. Autophosphorylation occurs in cis. A blue oval indicates the ABCDE domain; a P in white on a pink circle indicates the phosphate group.

## Data Availability

Data sharing is not applicable to this article.

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
