# Peer review of "Autophosphorylation and Self-Activation of DNA-Dependent Protein Kinase"

_genes, 2021, doi:10.3390/genes12071091_

Round 1

Reviewer 1 Report

The manuscript “Autophosphorylation and self-activation of DNA-dependent protein kinase” by Aya Kurosawa summarized current knowledge on the function of DNA-dependent protein kinase catalytic subunit (DNA-PKcs). The manuscript is useful for broad range of readers. I recommend this paper for the publication. Here are minor comments. Including the following discussions will improve the manuscript.

  1. (Essential function in adoptive immune system) One of the most important functions of DNA-PKcs is the activation of Artemis, which possesses hairpin opening during V(D)J recombination and DNA end trimming in NHEJ.

  1. (Evolutionary aspects) DNA-PKcs is not encoded in yeast, plants, and invertebrates, and the MRE11/RAD50/XRS2 complex appears to be critical for some of the DNA end resection in those organisms.

Author Response

Comments from Reviewer 1

The manuscript “Autophosphorylation and self-activation of DNA-dependent protein kinase” by Aya Kurosawa summarized current knowledge on the function of DNA- dependent protein kinase catalytic subunit (DNA-PKcs). The manuscript is useful for broad range of readers. I recommend this paper for the publication. Here are minor comments. Including the following discussions will improve the manuscript.

Thank you very much for the favorable comments.

  1. (Essential function in adoptive immune system) One of the most important functions of DNA-PKcs is the activation of Artemis, which possesses hairpin opening during V(D)J recombination and DNA end trimming in NHEJ.

Thank you for your suggestion. I have added a new paragraph explaining the function of DNA-PKcs in V(D)J recombination (Lines 57-65, Page 2).

  1. (Evolutionary aspects) DNA-PKcs is not encoded in yeast, plants, and invertebrates, and the MRE11/RAD50/XRS2 complex appears to be critical for some of the DNA end resection in those organisms.

I appreciate the reviewer’s insightful comments. In the revised manuscript, I have mentioned a new paper related to organisms lacking DNA-PKcs. The paper by Lees-Miller et al. showed that while many representative model organisms lack DNA-PKcs, many do have a putative gene encoding DNA-PKcs (Prog Biophys Mol Biol. 2021 163:87). Based on this paper, I added the sentence “DNA-PKcs appears to have been lost or its functions may have been diversified as organ-isms adapted to their environment during the course of evolution” (Line 40, Page 1). I have also cited three papers related to the involvement of the MRE11/RAD50/XRS2 complex in DNA end resection, and added the following sentences “In organisms that lack the gene encoding for DNA-PKcs, such as fission yeast (Schizosaccharomyces pombe), the Mre11/Rad50/Xrs2 complex interacts with NHEJ factors such as DNA ligase IV, is involved in the end-resection step of NHEJ, and ensures NHEJ fidelity” (Lines 49-52, Page 2).

Reviewer 2 Report

The role of DNA-PKcs kinase activity is central to the NHEJ pathway. The autophosphorylation of DNAPK is essential. Using structural data, this review is focused on the autophosphorylation of this kinase. This synthesis work is well done but several points have to be completed. 
1- It is important to describe all autophosphorylations of DNApk in particular in response to DNA damage. The ABCDE and PQR clusters are well known, but there are other sites of DSB-induced autophosphorylations. For example S56 in cluster N, T946 and S1004 in cluster JK (see for example Neal et al 2011). A figure could be added. 
2- Author should describe the impact of signaling pathways on this kinase activity. Indeed some kinases act directly on the autophosphorylation of DNAPKcs including for example AKt1, EGFR which are often involved in cancer and are described to modulate the DNA-PKcs autophosphorylation (see Goodwin and Knudsen 2014).
3- In the same way the author should explain the importance of the cell cycle in particular the autophosphorylation of DNAPK during mitosis 
4- Author should describe the phosphorylation/OglcNacylation balance involved in the regulation of DNAPKcs autophosphorylation in response to DNA damage (Lafont et al 2020)

Author Response

Comments from Reviewer 2

1- It is important to describe all autophosphorylations of DNApk in particular in response to DNA damage. The ABCDE and PQR clusters are well known, but there are other sites of DSB-induced autophosphorylations. For example S56 in cluster N, T946 and S1004 in cluster JK (see for example Neal et al 2011). A figure could be added.

I sincerely appreciate the referee's detailed comments and agree with your comments. I have cited the paper by Neal et al., and the figure has also been corrected (Lines 152-164, Page 4 and Figure 2).

2- the impact of signaling pathways on this kinase activity. Indeed some kinases act directly on the autophosphorylation of DNAPKcs including for example AKt1, EGFR which are often involved in cancer and are described to modulate the DNA-PKcs autophosphorylation (see Goodwin and Knudsen 2014).

I appreciate these thoughtful and insightful comments. I have cited the paper by Goodwin and Knudsen to explain that AKT1 and/or EGFR promote the activation of DNA-PKcs in the revised manuscript (Lines 44-46, Pages 1-2).

3- In the same way the author should explain the importance of the cell cycle in particular the autophosphorylation of DNAPK during mitosis.

I thank the reviewer for his/her insightful comments. In the revised manuscript, I cited a paper by Jette and Lees-Miller (Prog Biophys Mol Biol 2015, 117:194), and changed the sentence to “For example, it has been reported that autophosphorylation of DNA-PKcs is involved in end-processing, pathway choice for DSB repair, and normal mitotic progression ” (Line 69-71, Page 2).

4- Author should describe the phosphorylation/OglcNacylation balance involved in the regulation of DNAPKcs autophosphorylation in response to DNA damage (Lafont et al 2020)

I appreciate these thoughtful and insightful comments. In the revised manuscript, I cited the paper by Lafont et al. to describe that “It has been reported that the autophosphorylation efficiency of S2056 is related to the balance with O-linked β-N-acetylglucosamine (O-GlcNAc) modification, a post-translational modification of proteins” (Lines 119-121, Page3). As I found that interaction of DNA-PKcs with casein kinase II and/or Akt1 also affects the autophosphorylation efficiency of S2056, I have also cited the paper by Jette and Lees-Miller. Once again, I sincerely appreciate the referee's comments.